# Finite Element Analysis and Parametric Study of Spudcan Footing Geometries Penetrating Clay Near Existing Footprints

**Long Yu \*, Heyue Zhang, Jing Li and Xian Wang**

State Key Laboratory of Coastal and Offshore Engineering, Dalian University of Technology, Dalian 116024, China; zhangheyue@mail.dlut.edu.cn (H.Z.); ljj28@mail.dlut.edu.cn (J.L.); ljj2828@mail.dlut.edu.cn (X.W.)

\* Correspondence: longyu@dlut.edu.cn; Tel.: +86 411 84706479

**Abstract:** Most existing research on the stability of spudcans during reinstallation nearing footprints is based on centrifuge tests and theoretical analyses. In this study, the reinstallation of the flat base footing, fusimform spudcan footing and skirted footing near existing footprints are simulated using the coupled Eulerian–Lagrangian (CEL) method. The effects of footprints' geometry, reinstallation eccentricity ($0.25D$–$2.0D$) and the roughness between spudcan and soil on the profiles of the vertical force, horizontal force and bending moment are discussed. The results show that the friction condition of the soil–footing interface has a significant effect on $H$ profile but much less effect on $M$ profile. The eccentricity ratio is a key factor to evaluate the $H$ and $M$. The results show that the geometry shape of the footing also has certain effects on the $V$, $H$, and $M$ profiles. The flat base footing gives the lowest peak value in $H$ but largest in $M$, and the performances of the fusiform spudcan footing and the skirted footing are similar. From the view of the resultant forces, the skirted footing shows a certain potential in resisting the damage during reinstallation near existing footprints by comparing with commonly used fusiform spudcan footings. The bending moments on the leg–hull connection section of different leg length at certain offset distances are discussed.

**Keywords:** spudcan; skirted footing; footprint; jack-up; clay; large deformation analysis

## 1. Introduction

### 1.1. Background

Jack-up units are self-elevating mobile platforms which are used extensively in the offshore oil and gas industry. A typical jack-up consists of a floatable hull and three independent retractable legs. The legs rest on spudcan footings that are usually circular or polygonal in plan and with an inverse cone underneath. Once a jack-up unit is towed to site, its installation begins by lowering the legs to the seabed and pushing the spudcans into the soil and then rising the hull over the water. Then pre-loading can be achieved by pumping water into the hull. The pre-loading makes the spudcan penetrate deeper to provide more resistance. After pre-loading, the water is pumped out and the spudcan's bearing capacity has some reservation. After all the work of the jack-up has finished, it is removed from the site by retracting the legs from the seabed. The processes of installation and extraction of the spudcan leave a permanent seabed depression at each footing site, which is referred to as a "footprint".

The footprint changes the seabed in two ways, as shown in Figure 1: An inclined seabed surface and a varying soil strength profile within the footprint (normally decreasing soil strength due to remolding). Both of them result in additional horizontal forces and bending moments compared with the initial installation. The spudcan–footprint interaction problem is significant as it can lead to significant time loss, cost implications, risks to adjacent structures and potential injury to personnel. Dier et al.

concluded from industry practice data that incidents caused by uneven seabed/scour/footprint are at a rate of 15% of the total [1]. This rate has increased obviously due to increasing demands of jack-up operation close to previous sites in recent years [2].

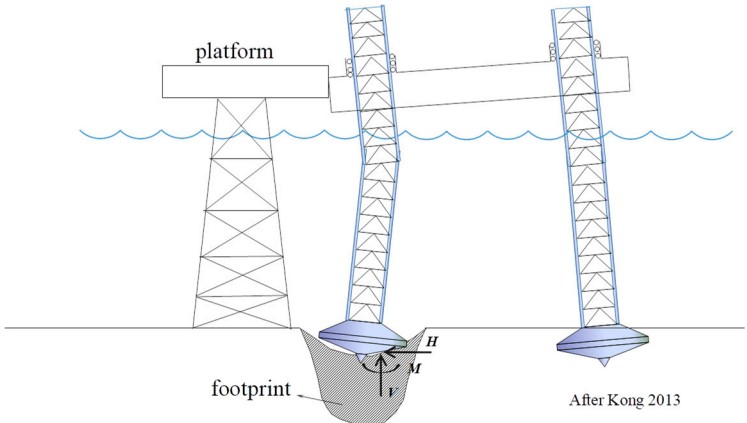

**Figure 1.** The failure mechanism of spudcan reinstallation near a footprint.

## *1.2. Previous Work*

The problem of jack-up reinstallation near the existing footprints attracted more attention in the recent 10–15 years. In some studies [3,4], footprints resemble an inverted conical shape cavity. The installation, operation, and removal of the spudcan can also remold the surrounding soil, resulting in highly variable shear strength profiles in the vicinity of the footprints [4–8].

Hartono et al. [9] used an experimental method (centrifuge tests) and numerical analysis (simulated with ABAQUS/CEL) respectively to investigate the efficacy of reaming technique in mitigating the footprint hazards. He found that the numerical results demonstrate good agreement with experiment results and reaming can be a viable option to mitigate spudcan–footprint interaction. He strongly suggested making numerical modeling as a viable tool for site-specific assessment of spudcan–footprint interaction problem. Like Hartono, the CEL large deformation method is adopted in this study to investigate the reinstallation behaviors of flat base footing, fusiform spudcan footing, and skirted footing.

Spudcans are the most common footings used for jack-up units. Along with the improvement of technology and the increasing demands of operating on the very soft soils, the footings become larger in diameter and flatter at the base. The geometries of typical fusiform spudcan footings are shown in Figure 2. The investigations from some research shows that, by comparing with fusiform spudcan footing, skirted footing may have a higher bearing capacity [7] and have some potential in mitigating punch-through failure [10,11].

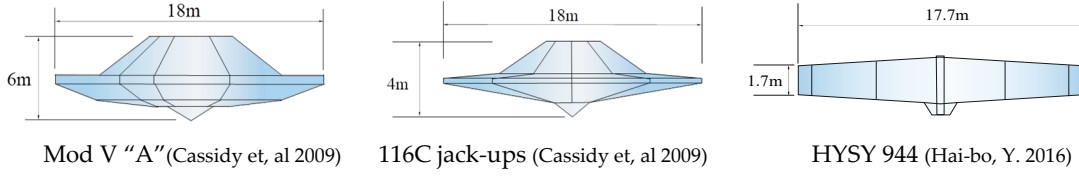

**Figure 2.** Three typical spudcans [3,12].

Cassidy et al. [3] used a 1:250 scale model of current Mod 'V 'jack-up in their centrifuge tests to simulate the interaction between real spudcan and soil. Kong et al. [13,14] replaced real spudcan with flat base footing in their centrifuge tests to eliminate the variables related to spudcan geometry. Zhang et al. [15] simulated spudcan with flat base footing in his numerical study to make sure that the touchdown level of footing could be identified clearly.

Gan et al. [4,6,8] studied the spudcan–footprint interaction considering the 'real' initial penetration. Their research showed that the soil is obviously disturbed during the initial penetration and will recover with time. To simplify the problem, many of the following studies assumed an artificial footprint, such as Kong [13], Zhang et al. [15], Jun et al. [16–19]. The assumption of an artificial reverse cone footprint may respond to a fully recovered 'real' footprint after a long period from the initial penetration. Therefore, the three idealized footprints TA, TB, and TC following Kong [13] are adopted in this study to simplify the numerical model.

The distance from the central line of the footprint to the reinstalled spudcan center was termed as offset distance or reinstallation eccentricity ($\beta$), which was proved to be a key issue to the profiles of bending moment ($M$) and horizontal force ($H$). Stewart [20] carried out centrifuge model tests and the results showed that both $M$ and $H$ increased to an obvious value when $\beta/D = 0.5$ to 1.0, where $D$ is the diameter of the reinstalled spudcan, and $H$ reached to the maximum value when $\beta/D = 0.75$. Cassidy et al. [3] founded that $M$ and $H$ were most obvious when $\beta/D = 0.5$ and became very small when $\beta/D > 1.5$. Carrington [21] carried out large deformation numerical analyses to simulate the reinstallation processes with $\beta/D = 0.167$ to 0.407, and obtained a most critical case at $\beta/D = 0.29$. Kong et al. [13,14]. Investigated the effect of footprints with various size and slope angles. In their study the critical case was $\beta/D = 1.0$.

Some research showed that the fixity condition at the leg–hull connecting point has a significant effect on the reinstallation behavior near a footprint [3,5,22–25]. It can be concluded that harder fixity tends to increase the maximum value of $M$ and $H$ but reducing the lateral movement of the spudcan during reinstallation.

### 1.3. Motivation of Present Study

Most existing research on the stability of spudcans during reinstallation nearing footprints is from centrifuge tests and theoretical analyses. In this study, the reinstallation of flat base footing, fusiform spudcan footing and skirted footing near existing footprints are simulated using the coupled Eulerian–Lagrangian (CEL) method. The effects of footprints' geometry, reinstallation eccentricity and the roughness between spudcan and soil on the profiles of vertical force, horizontal force and bending moment are discussed. One purpose of this study is to reveal the mechanisms of those factors which affect $V$, $H$, and $M$ profiles during reinstallation, by presenting the soil flow mechanisms of selected cases.

The other purpose is to discuss the effect of footing geometry shape during reinstallation near existing footprints. Flat base footings, fusiform spudcan footings, and skirted footings are investigated in this paper. Fusiform spudcan footings have been widely used in practice. Skirted footings have been proved to have some potential in bearing capacity and mitigating punch-through failure, but its behavior in mitigating footprint hazards is still not very clear. Flat base footings have advantages in eliminating the uncertainty when discussing the soil flow mechanism, by comparing with fusiform spudcan footings of which the reverse cone initially touches the seabed. Besides, some large footings in practical engineering have a relatively flat base, such as HYSY 944, as shown in Figure 2.

## 2. Materials and Methods

### 2.1. Modeling of Footings

The sign convention and definition of terminology in this study are plotted in Figure 3 and the numerical models of footings investigated in this study are shown in Figure 4.

The diameter of all the footings is $D = 15$ m and the height of the max area is $H_t = 1.75$ m. The geometry of the spudcan follows Liu et al. [26] and Yu et al. [27]. The geometry of the skirted footing is $H_s = 0.25D = 3.75$ m and $T_s = 1.75$ m; where $H_s$ is the height of the skirt and $T_s$ is the thickness of the skirt. The geometry of the fusiform spudcan footing is $D = 15$ m, $H_1 = 2.5$ m and $H_2 = 3.3$ m. The distance from the center of Section 1.1 to the reference point (RP) for flat base footing is $H_a = 1.75$ m,

for fusiform spudcan footing is $H_a$ = 7.55 m, for skirted footing is $H_a$ = 5.5 m. To simplify the problem, the leg and footing are constrained as rigid.

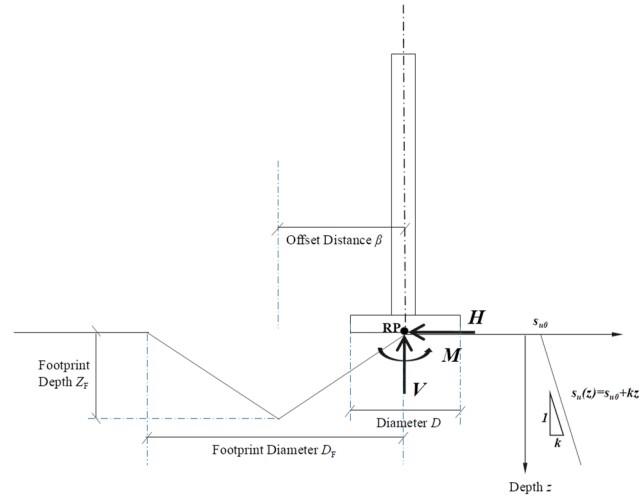

**Figure 3.** Sign convention and definition of terminology.

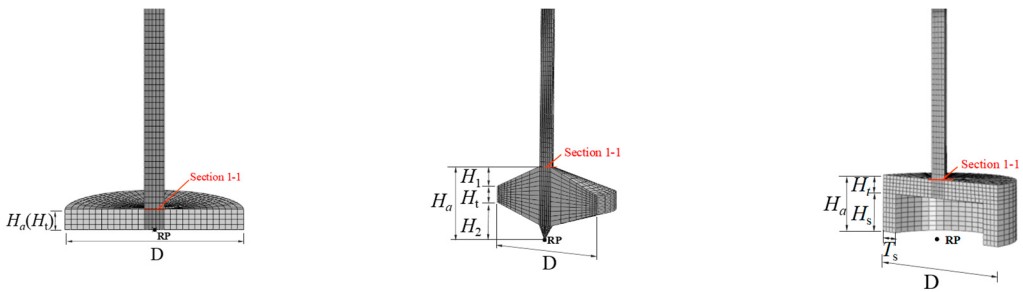

**Figure 4.** The dimensions of footings: flat base footing, fusiform spudcan footing and skirted footing.

### 2.2. Model of Soil

To simplify the problem, the footprint in this study is idealized as a reverse conical cave on the soil surface. The ideal elasto-plastic model is used to describe the stress–strain relationship of the soil, obeying the Mohr–Coulomb strength criterion. The undrained shear strength profile is $s_u$ = 7.5 + 0.92z kPa, where $z$ is the soil depth from the mudline. The Poisson's ratio is $v$ = 0.49. The elastic modulus is $E$ = 500$s_u$. The effective unit weight is $\gamma'$ = 6.82 kN/m³. The internal friction angle and the dilation angle are $\varphi$ = $\Psi$ = 0°. The load is achieved by displacement control at a rate of $v$ = 0.5 m/s, which is a compromise between the accuracy and the efficiency.

The principle of universal contact is used to simulate the contact property between footing and soil. In tangential direction, the penalty function is selected to model the friction condition, thus different frictions can be tested. In normal direction, "hard" contact is set to simulate the interface, which can transfer positive pressure without limitation but separate under tension.

A half model is modeled because of the symmetry. Both a cuboid soil domain and a cylindrical one have been tested. The results show that the former is more efficient and easier to mesh without the loss of accuracy. Thus, the cuboid soil domain, as shown in Figure 5, is used in this study. The soil is modeled by EC3D8R element (three-dimensional, eight-node linear brick, multimaterial, reduced integration with hourglass control) and the footings are modeled by C3D8R (three-dimensional, eight-node linear brick, reduced integration with hourglass control) element in ABAQUS/Explicit. In order to eliminate the influence of boundary effect, the width, depth, and thickness of the soil are $8D$, $4D$, and $4D$, respectively. In addition, there is an empty element layer, 4 m thick, at the top of the soil to heave up during reinstallation. The mesh close to the footing penetrating path is refined.

The minimum element size is $d_{\min}/D = 1/30 = 0.5$ m. The mesh density and soil domain have been proved to be with acceptable accuracy.

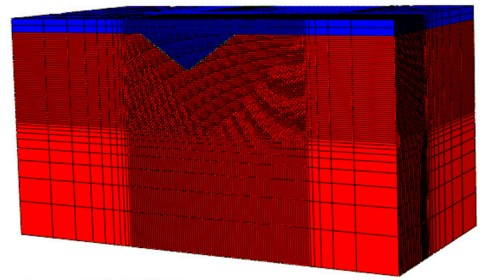

**Figure 5.** The mesh of soil.

*2.3. Numerical Cases*

In this study, the reinstallation process of three 15 m diameter footings with offset distances of $\beta/D = 0.25$, 0.5, 0.75, 1.0, 1.25, 1.5, and 2.0 are simulated respectively. Four types of footprints (one of them is a flat surface field) are simulated, as listed in Figure 6. TA, TB, and TC are three footprints with various depths, and FS means flat surface (no footprint). The naming rule of each case is similar to that of Kong et al. [13]. For example, TB-2*D*-0.25*D* means the footprint is TB type with a diameter of $D_F = 2D$, and the eccentricity of reinstallation is $\beta = 0.25D$. All the cases investigated in the paper are listed in Table 1.

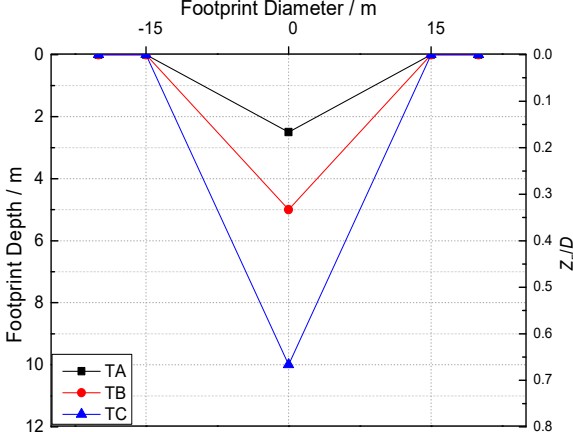

**Figure 6.** Dimensions of three types of footprint.

**Table 1.** Numerical cases.

| Footprint Type | Prototype (m) | | $\Theta$ (°) | $\beta$ (m) | Case Name |
|---|---|---|---|---|---|
| | $D_F$ | $Z_F$ | | | |
| TA | 30 | 2.5 | 9.5 | 0.25*D*, 0.5*D*, | TB-2*D*-0.25*D*(μ) |
| TB | 30 | 5 | 18.4 | 0.75*D*, 1.0*D*, | (μ is friction |
| TC | 30 | 10 | 33.7 | 1.25*D*, 1.5*D*, | coefficient) |
| Flat surface | - | - | - | 2.0*D* | FS |

## 3. Results

*3.1. Effect of An Existing Footprint*

At first, the flat base footing is taken as an example to show how an existing footprint affects the resistance profile during installation and to show the soil flow mechanism. Zero depth is defined as

the maximum cross-section area of footing touching seabed level. *V*, *H*, and *M* denote the vertical force, horizontal force, and bending moment acting on the reference point, respectively.

The *V*, *H*, and *M* profiles are plotted in Figure 7 and the soil flow mechanisms are shown in Figure 8. The positive *H* value means a horizontal force towards the footprint. The positive *M* value means an anti-clockwise bending moment acting on the reference point (RP).

In this case, the horizontal force comes from two parts: (1) When $z/D < 0.15$, with the penetration goes by, the soil under the footing is pushed into the footprint while the footing is constrained without horizontal movements. The relative motion provides a friction force towards the footprint (as the yellow arrow shown in Figure 8a. When $z/D < 0.15$, the maintained zero $H/AS_u$ value of case TB-2*D*-1.0*D* (smooth) as shown in Figure 7a confirms this conclusion. (2) As the footing penetrates deeper ($z/D > 0.15$), the soil on the footing's right side is compressed, which provides a leftward earth pressure (as the green arrows shown in Figure 8c. When the footing reaches the toe of the footprint, 0.33*D*, the soil on the footing's left side heaves up and provides a rightward earth pressure. After that, the total horizontal force reduces. When the penetration depth reaches ~0.8*D*, the soil on both the left and right sides show a symmetric fully flowing back mechanism, as shown in Figure 8d. The symmetric soil flow mechanism results in that the *H* and *M* values are much smaller.

*M* peaks as soon as the footing touches the seabed (around $z/D = 0.02$). This is because that at this depth the eccentric distance of the resultant vertical resistance force is very large, as shown in Figure 8a, although the vertical resistance is far from the peak value at this depth. As the penetration depth increases, the eccentric distance of vertical force reduces and as a result the corresponding bending moment obviously reduces.

Compared with the centrifuge test results from Kong [25], the horizontal force profile of Kong's lies between the smooth and rough cases of this study (Figure 7a) because the friction characteristic of the centrifuge test on the interface of aluminum footing and the soil is between rough and smooth. It can be seen that the numerical results of this study and the centrifuge test results from Kong [25] have very similar *H* and *M* profile trends.

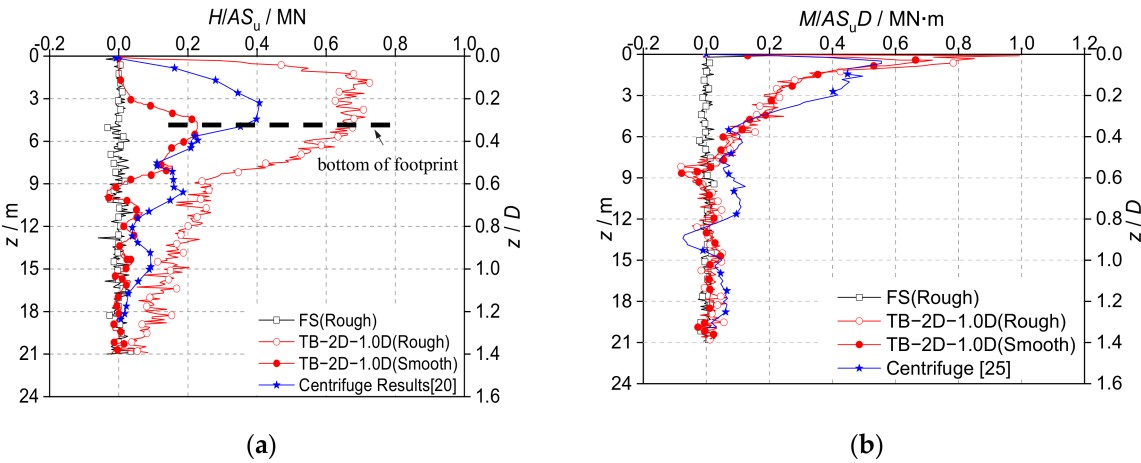

**Figure 7.** Effect of existing footprint on the *V*, *H*, and *M* responses of the flat base footing.

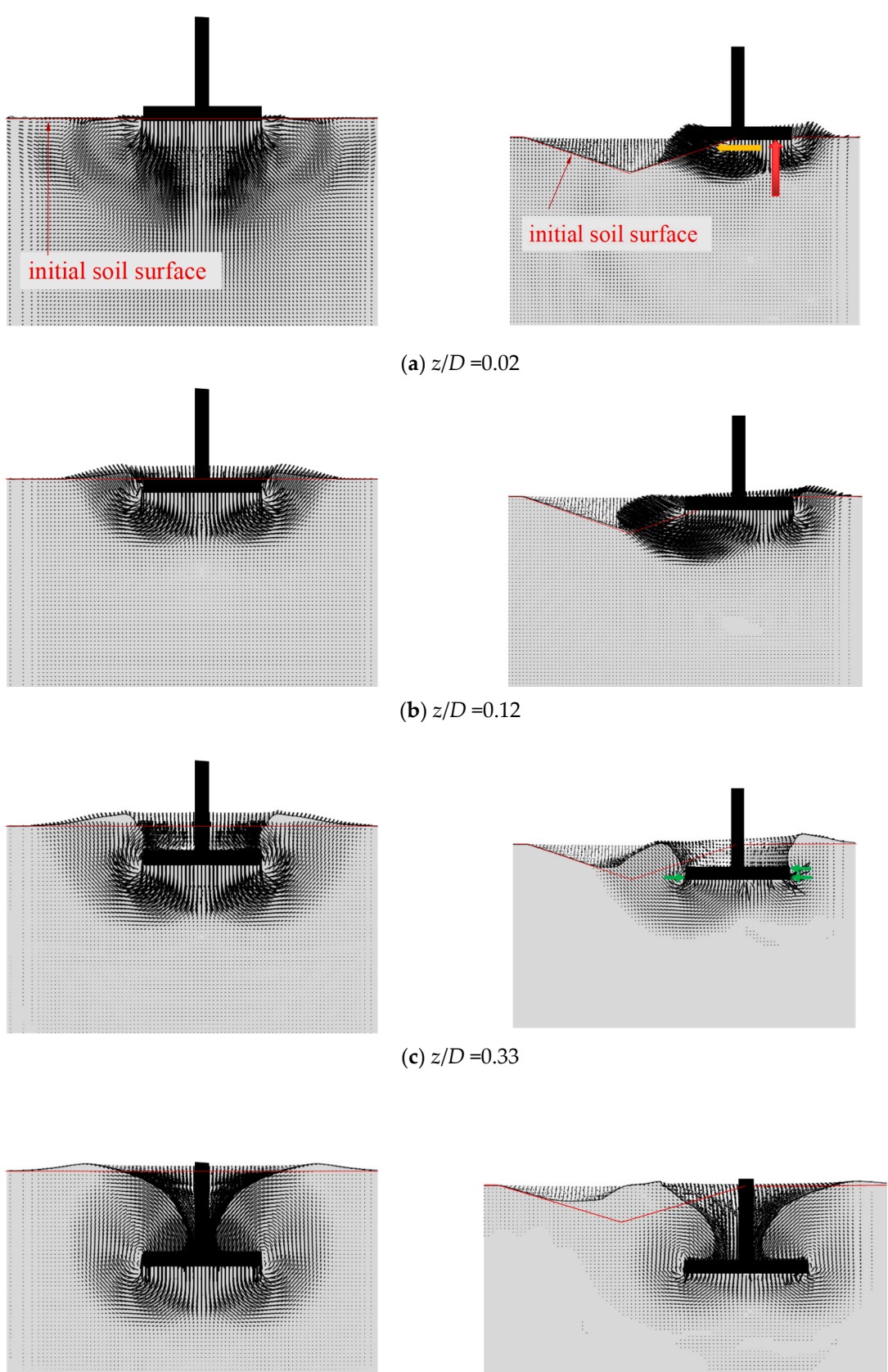

**Figure 8.** Soil flow mechanism of cases FS (**left**) and TB-2*D*-1.0*D* (**right**) for flat base footing.

### 3.2. Effect of An Existing Footprint

In Abaqus/CEL, the basic coulomb friction model defines the maximum allowable friction (shear) stress across an interface to the contact pressure stress, $\tau_{max}$, as a function of the contact pressure:

$$\tau_{max} = \mu\, p \tag{1}$$

in which $p$ is the contact pressure and $\mu$ is a friction coefficient that can be any non-negative value. In some special cases, the contact pressure $p$ might be so large that $\tau_{max} = \mu\, p$ exceeds the yield stress in the material beneath the contact surface, thus a shearing limit value, $\tau_{limit}$, is adopted to avoid this situation. Regardless of the magnitude of the contact pressure stress, sliding will occur if the magnitude of the equivalent shear stress reaches $\tau_{limit}$. When both $\tau_{max}$ and $\tau_{limit}$ exceed $s_u$ (yield stress), the maximum allowable friction (shear) stress equals $s_u$. All in all, the $\mu$ value only affects the friction force before the contact pressure reaches $p_2 = s_u/\mu$. After that, the friction force would be equal to $\sim s_u$ due to the yielding of clay. The relationship between equivalent shear stress and the contact pressure is plotted in Figure 9.

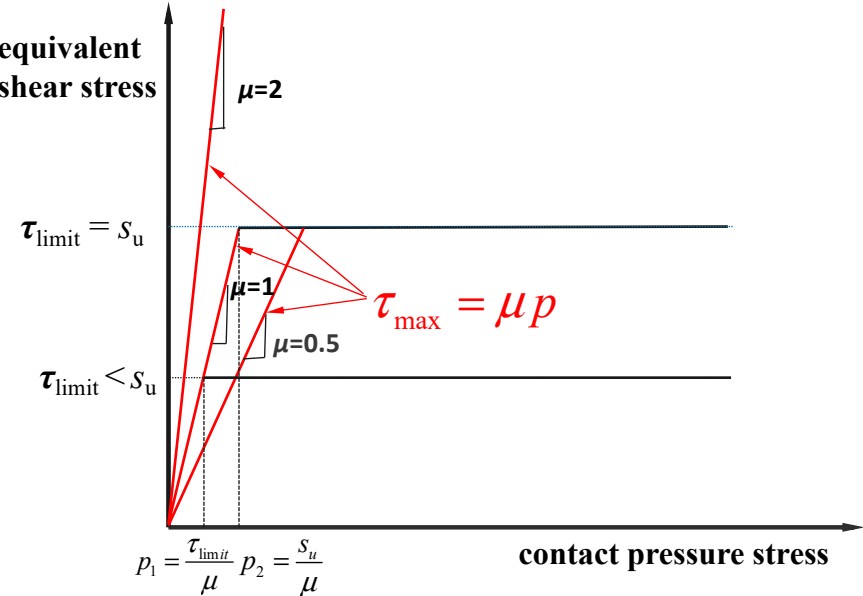

**Figure 9.** Behavior of the contact element in Abaqus/CEL.

A simple test model, as shown in Figure 10, is created to verify the accuracy of calculating friction force by Abaqus. All the parameters are detailed in Figure 10. The three anchors are disconnected and go right at a speed of 0.1 m/s at the same time. Only the friction force of anchor 2 on the contact surface surf 2 are considered, anchor 1 and anchor 3 are created to eliminate the influence of back-flow soil. An empty element layer of 5 m thick at the top of the soil is set to allow soil heaving.

In the simple test model, the friction coefficient is set to $\mu = 10,000$ and the shearing limit value is set to $\tau_{limit} = 5.5$ kPa (larger than $s_u = 5$ kPa). According to Equation (1), $\tau_{max} = \mu p = 10,000 \times 70 = 700$ MPa, which is far greater than $\tau_{limit}$. The cases with different mesh size and calculated results are listed in Table 2 and plotted in Figure 11. It can be seen that the calculated friction force is a little lower than the theoretical solution, which may be because of the fractional volume method in CEL. The numerical friction force is getting close to the theoretical solution as the mesh density increases. When the minimum element size is $b_{min}/B = 1/30$, the calculation error is 6%, which is selected in the following analyses.

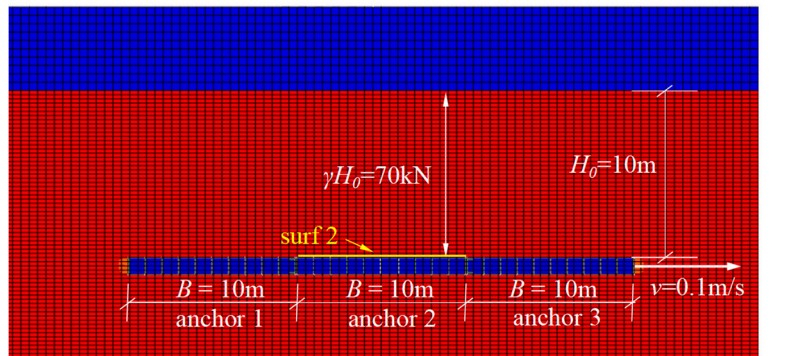

**Figure 10.** The test model.

**Table 2.** Numerical cases and results.

| The minimum Element Size $b_{min}/B$ | Numerical Friction Force (kPa) | Theoretical Friction Force (kPa) | Calculation Error (%) |
|---|---|---|---|
| 1/20 | 4.55 | 5 | 9 |
| 1/30 | 4.7 | 5 | 6 |
| 1/40 | 4.75 | 5 | 5 |
| 1/80 | 4.85 | 5 | 3 |

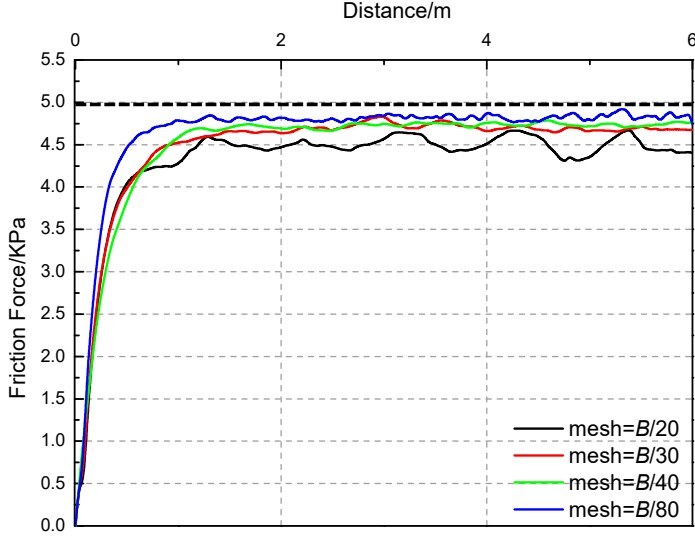

**Figure 11.** Numerical friction force in test model.

After investigating the behavior of the friction element in CEL, the effects of soil–structure friction on the *V H* and *M* of a spudcan penetrating near an existing footprint are carried out. The friction coefficient is set to $\mu = 0$ and 10,000 to represent smooth and rough conditions respectively. The shearing limit value is set as the undrained shear strength of the surrounding clay.

Comparing the smooth and rough cases, it can be seen that some certain friction has a significant effect on *H* profile, but no obvious effects on *V* and *M*, as shown in Figure 12. The friction condition does not affect the location where $H_{max}$ and $M_{max}$ occur. The soil flow mechanism in Figure 13 explains how the friction affects *H* profile. For the smooth case, *H* is only from the lateral pushing force of the soil on the right side of the footing. While for the rough case, the friction on the footing bottom also contributes to *H*.

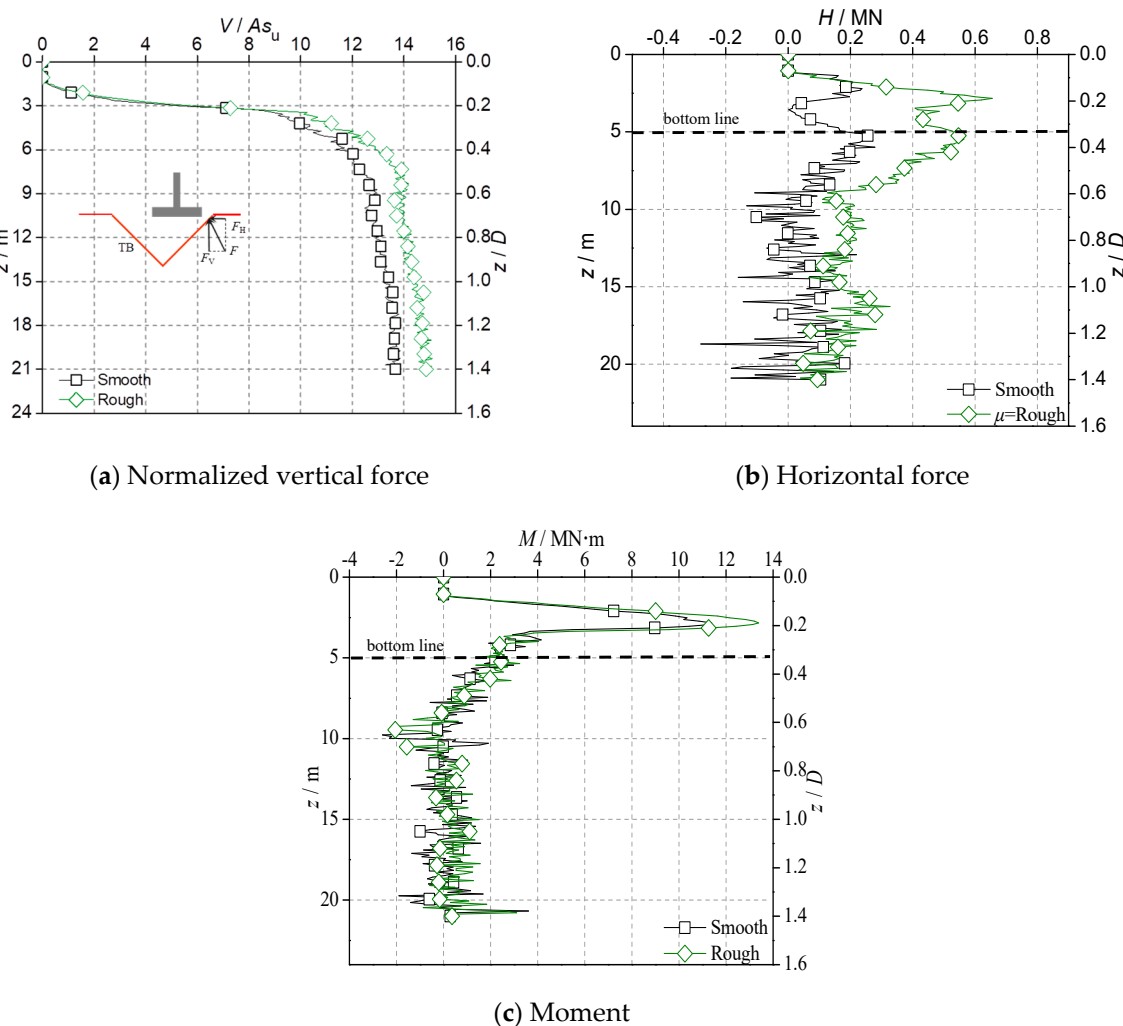

(**a**) Normalized vertical force      (**b**) Horizontal force

(**c**) Moment

**Figure 12.** The *V*, *H*, and *M* profile during smooth and rough conditions. (TB−2*D*−0.25*D*).

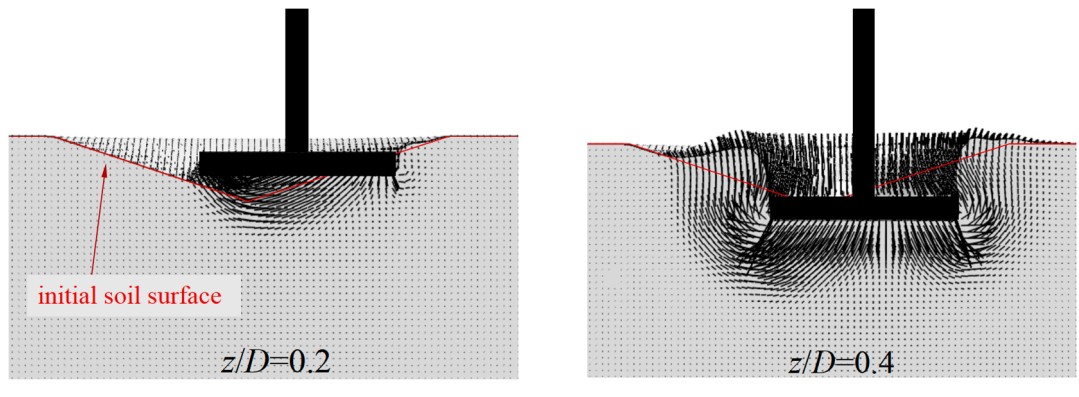

(**a**) smooth

**Figure 13.** *Cont.*

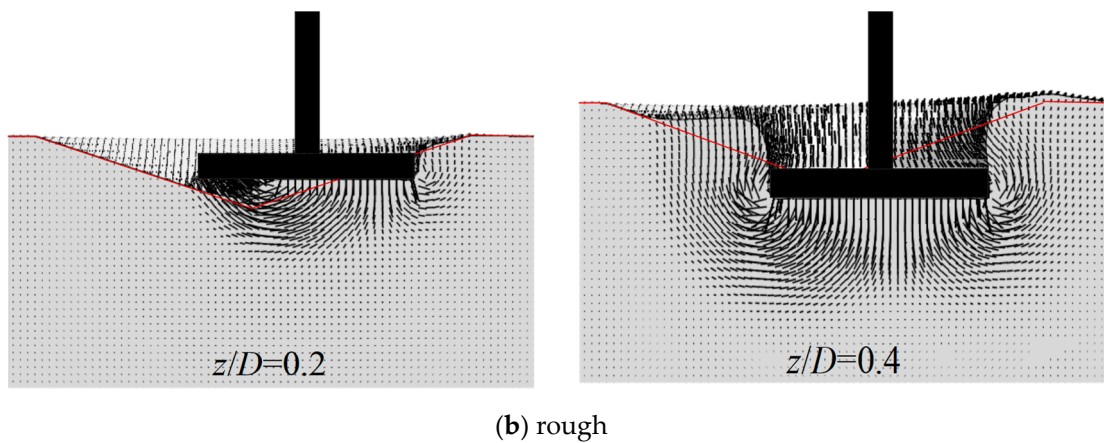

(**b**) rough

**Figure 13.** The soil flow mechanism of (**a**) smooth and (**b**) rough conditions. (TB−2*D*−0.25*D*).

The maximum normalized values of *H* and *M* of flat base footing are summarized in Figure 14. It can be seen clearly that the maximum *H* value of rough cases is around three times of that of smooth case, while the friction condition has a much smaller effect on $M_{max}$ values (increasing 1.2 to 1.4 times).

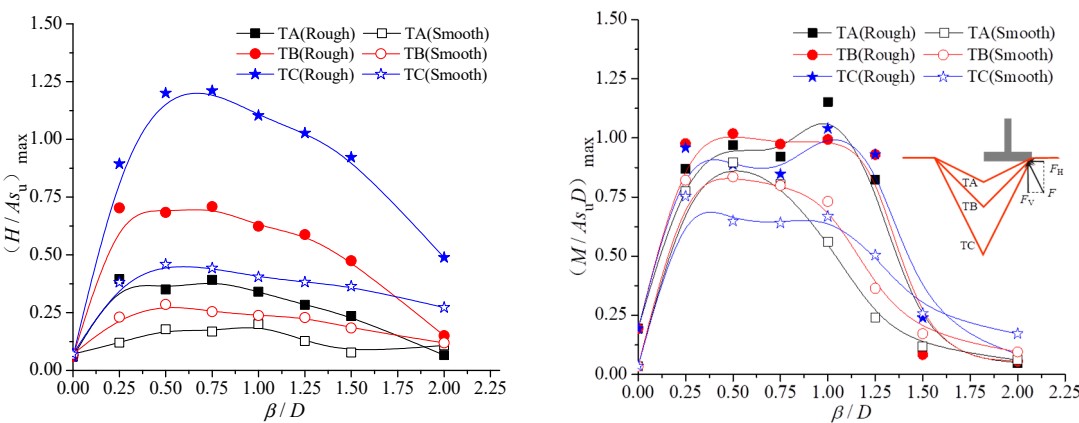

**Figure 14.** Maximum normalized *H* and *M* values against eccentricity ratio (flat base footing).

### 3.3. Effect of the Location of the Reference Point (Working Leg Length)

For the convenience of discussion, the *V*, *H*, and *M* discussed above are obtained using a reference point (see RP in Figure 4). If the reference point is located at the leg–hull connection section (see $RP_0$ in Figure 15), an additional moment ($M_a$) will be mobilized by *H* and its eccentricity (i.e., the working leg length), while the horizontal and vertical forces are not affected by the location of RP. In practical engineering cases, the leg–hull connector section could be the most dangerous section.

Assuming that the top head of the leg is fully fixed and the footing is considered as a rigid body, the additional bending moment ($M_a$) at $RP_0$ due to the horizontal force acting on the footing can be calculated as $M_a = H * L_{w\text{-}leg}$. The bending moment at $RP_0$ ($M_{hull}$) varies with the working leg length. The total moment at $RP_0$ ($M_{hull}$) can, therefore, be calculated as $M_{hull} = M + M_a$. The maximum value of both horizontal force ($H_{max}$) and bending moment ($M_{max}$) are taken as the most unfavorable combination of loads to calculate the bending moment on the leg–hull connection at different working leg lengths. As an example, the profile of $M_{hull}$ of the flat base footing reinstalling near the TA footprint is plotted in Figure 16. It can be seen that $M_{hull}$ is within a positive value at a small leg length, which means an anticlockwise moment. With the increasing of the working leg length, $M_a$ increases linearly and, as a result, the total moment $M_{hull}$ decreases. When $L_{w\text{-}leg}$ is less than ~30 m, the total

moment is within a negative range (clockwise). With further increasing of $L_{\text{w-leg}}$, the absolute value of the clockwise $M_{\text{hull}}$ would be larger than the anticlockwise $M_{\text{hull}}$ at $L_{\text{w-leg}} = 0$.

Considering working leg length, the bending moment, $M_{\text{hull}}$, is a combination of the $H$ and $M$ at RP. To simplify the discussions, only the moments at the lower end of the leg (Section 1.1), $M_{1\text{-}1}$, are presented.

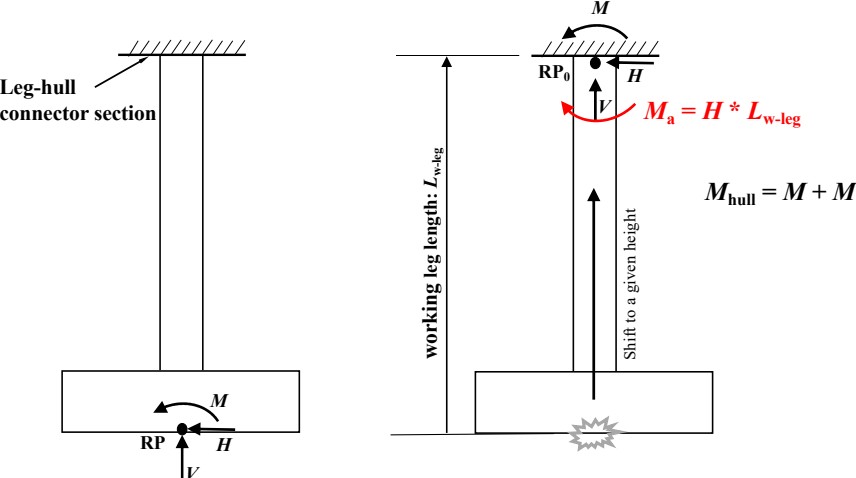

**Figure 15.** Maximum normalized $H$ and $M$ values against eccentricity ratio (flat base footing).

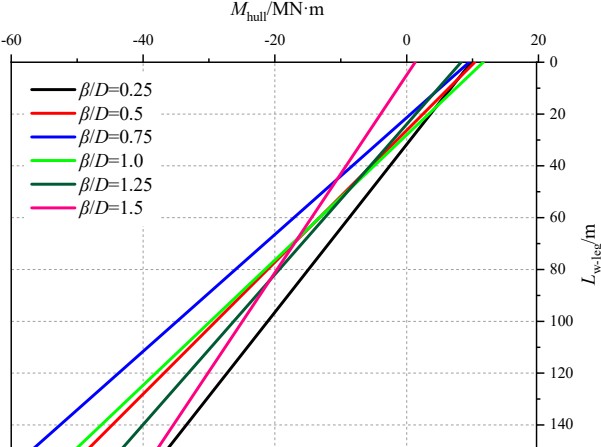

**Figure 16.** Maximum normalized $H$ and $M$ values against eccentricity ratio (flat base footing).

### 3.4. Effect of Footprint Geometry

The resistance profiles of spudcan penetrating through the edge of footprints TA, TB, and TC are presented in Figure 17, in which the offset distance is 0.75$D$.

As expected, the deeper the footprint is, the more effect it has on the reinstallation resistance profiles. All three $H$ Profiles have the same trend, but the case with steeper slope causes higher $H$ values. The $H_{\text{max}}$ value for TC is about 3–5 times higher than that for TA. The deeper the footprint is, the longer it takes for $H$ to reduce to zero.

The bending moment at Section 1.1 ($M_{1\text{-}1}$) can be derived according to the $V$, $H$, and $M$ values acting on the Reference Point. The vertical force acting on the RP has no contribution on the bending moment at the section, $M$ has a positive contribution, and $H$ times distance has a negative contribution. The maximum $M_{1\text{-}1}$ values occur at a very shallow depth. With further penetration, the horizontal force becomes larger and plays a leading role in $M_{1\text{-}1}$ value. This results in that the positive $M_{1\text{-}1}$ reduces gradually to negative in Figure 17, with an increasing penetration depth.

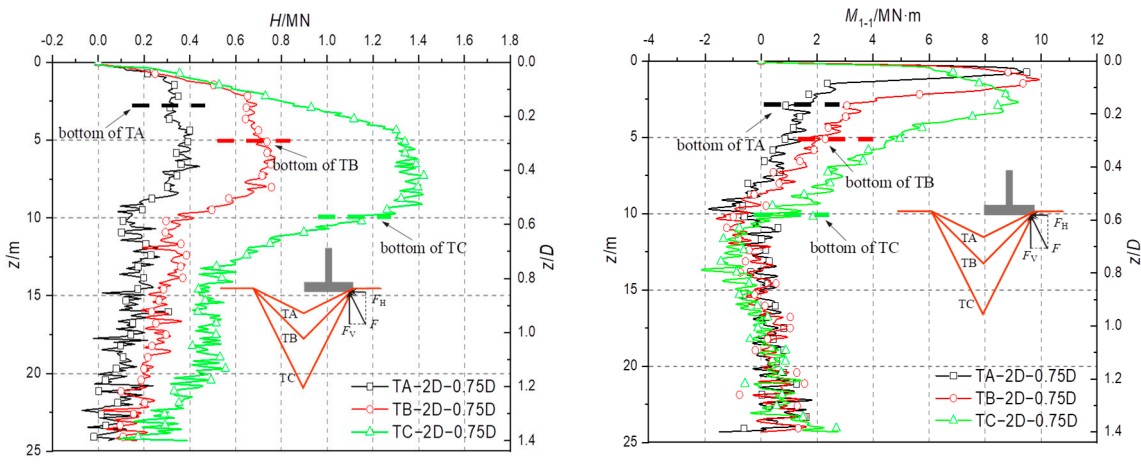

(**a**) Effect of footprint geometry on flat base footing (*β* = 0.75*D*)

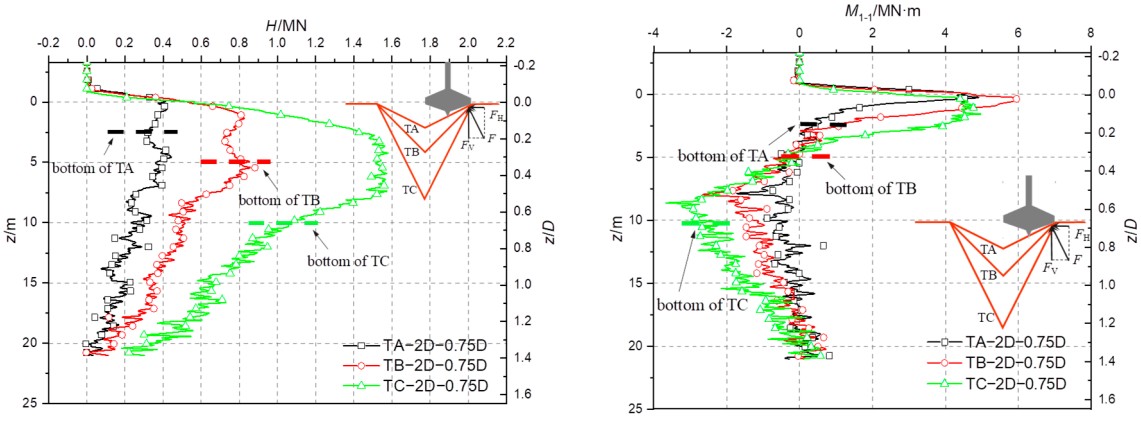

(**b**) Effect of footprint geometry on fusiform spudcan footing (*β* = 0.75*D*)

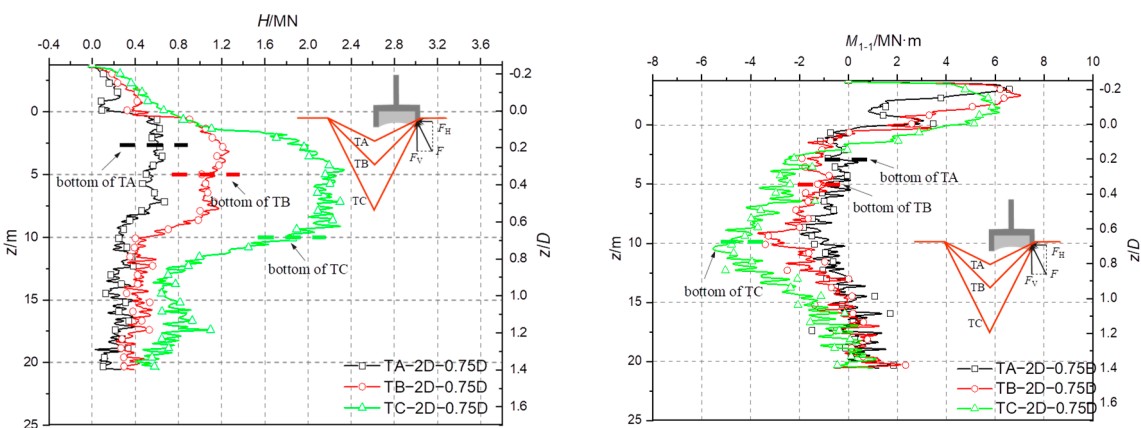

(**c**) Effect of footprint geometry on skirted footing (*β* = 0.75*D*)

**Figure 17.** Effect of footprint geometry on three types of footings.

*3.5. Effect of Footings' Geometry Shape and Offset Distance*

The *H* and *M* profiles of the three footings reinstalling at selected typical offset distances are shown in Figure 18. The *H* can be separated into two parts. The first is the horizontal component of normal contact force between the footprint slope and the right side of footing, which is the primary cause of the first peak shown in Figure 19. The second is the lateral pushing force from the right-side soil caused by

the asymmetry soil flowing, which is the primary cause of the second peak. For a flat base footing or a skirted footing, the horizontal component of normal contact force is relatively small, since the footing base is horizontal. However, for a fusiform spudcan footing, due to the inverted conical shape, the first part of $H$ force plays a leading role in $H$ profile. After deep penetration, the geometry shape has a minor effect on the resistance, since the soil flow mechanisms are both fully back flow left and right.

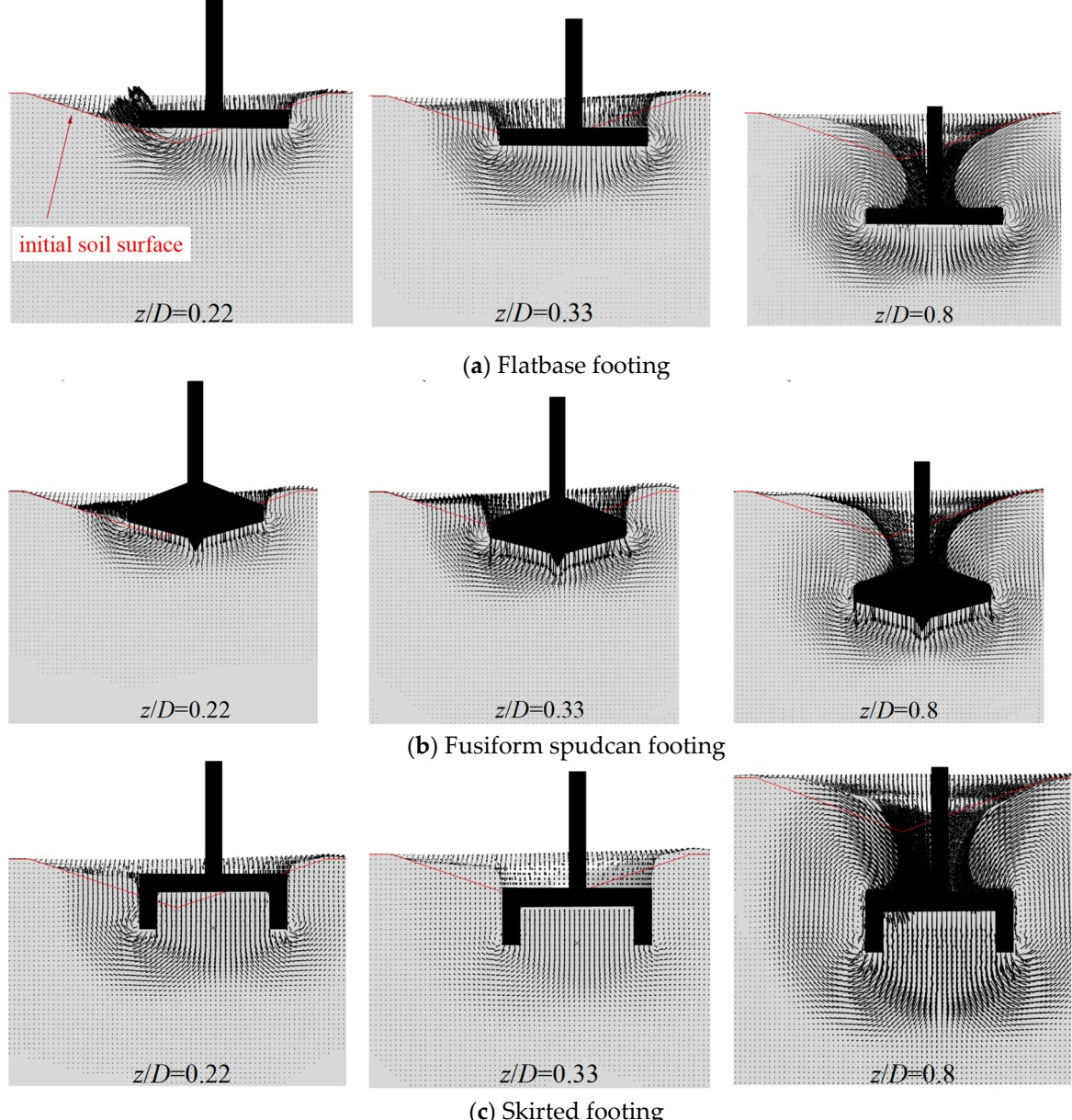

**Figure 18.** The soil flow mechanism for different footings (TB-2$D$-0.25$D$).

The $H_{max}$ and $M_{1\text{-}1max}$ values of all the cases in this study are listed in Table A1 in Appendix A and plotted in Figure 20. For flat base footings and skirted footings, both $H_{max}$ and $M_{1\text{-}1max}$ are significant when $\beta/D$ = 0.25 to 1.25. When $\beta/D \geq 1.5$, the value of $M_{1\text{-}1max}$ reduces to zero, while $H_{max}$ still remains at significant values. For fusiform spudcan footings, both $H_{max}$ and $M_{1\text{-}1max}$ are significant when $\beta/D$ = 0.25 to 0.5. From the perspective of the footing shape, the flat base footing gives the lowest $H_{max}$ but the largest $M_{1\text{-}1max}$, and the performances of the fusiform spudcan footing and the skirted footing are similar.

It is worthwhile to note that the thickness of the skirt for the skirted footing of the numerical model is higher than in situ skirted footing in order to mitigate numerical divergence. That might cause an overprediction on resistance loads. The effect of the skirt thickness can be ignored when the base level (with the maximum cross-section area) of the skirted footing fully touches the soil.

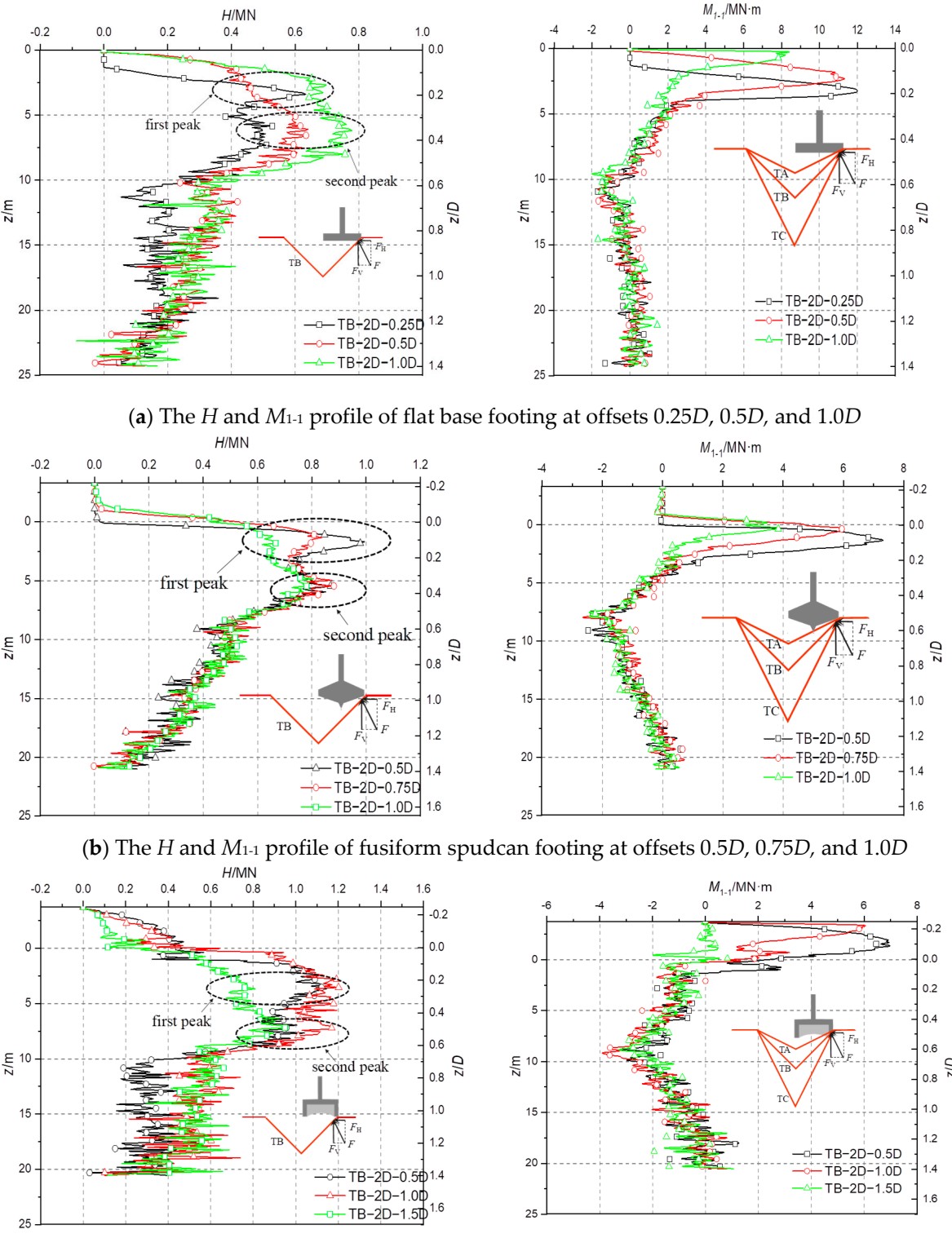

(**a**) The *H* and *M*₁₋₁ profile of flat base footing at offsets 0.25*D*, 0.5*D*, and 1.0*D*

(**b**) The *H* and *M*₁₋₁ profile of fusiform spudcan footing at offsets 0.5*D*, 0.75*D*, and 1.0*D*

(**c**) The *H* and *M* profile of skirted footing at offsets 0.5*D*, 1.0*D*, and 1.5*D*

**Figure 19.** The *H* and *M*₁₋₁ profile of three kind of footings at different offsets.

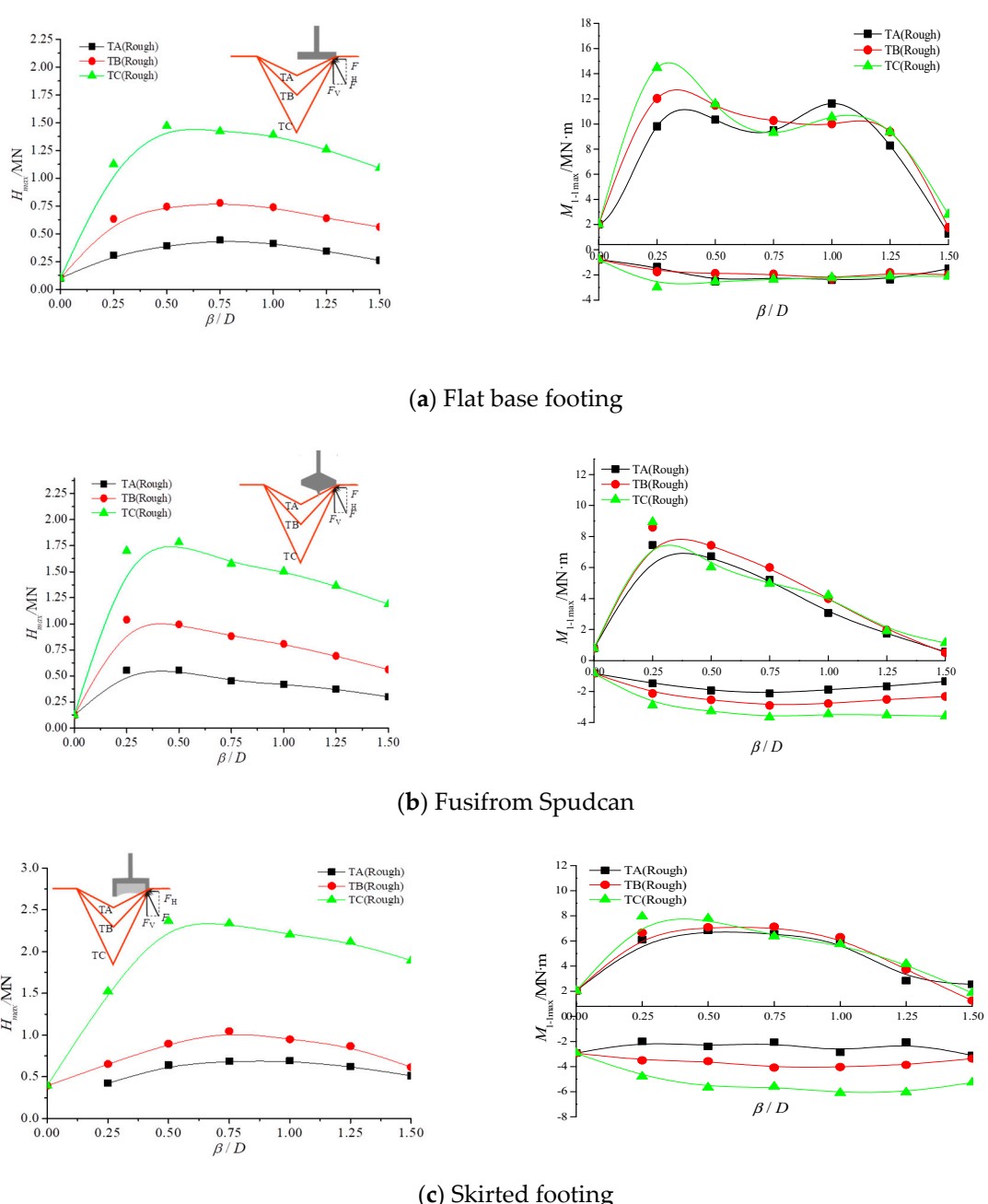

**(a)** Flat base footing

**(b)** Fusifrom Spudcan

**(c)** Skirted footing

**Figure 20.** Maximum values of *H* and *M* against eccentricity ratio.

### 3.6. Resultant Force of V H and M

The above analyses are based on *V H* and *M* values at the reference point. To provide another view, the *V, H,* and *M* values of each case can be transformed into one resultant force acting on a point at the footing base level. The resultant force has an inclination of $\alpha = \tan^{-1}(H/V)$ to the vertical line and an offset ratio of $e/D = M/VD$ to the central line of the footing. From Figure 21, it can be seen that when $z > 0$ m, the load inclination $\alpha$ and eccentricity $e/D$ of skirted footing is smaller than that of fusiform spudcan footing. When $z < 0$ m, although both $\alpha$ and $e/D$ of the skirted footing are larger, the vertical force is relatively small and the force acting on the footing may not be sufficient to cause structure failure. That is to say, the skirted footings may have a certain potential in resisting the damage during reinstallation near existing footprints, by comparing with commonly used fusiform spudcan footings.

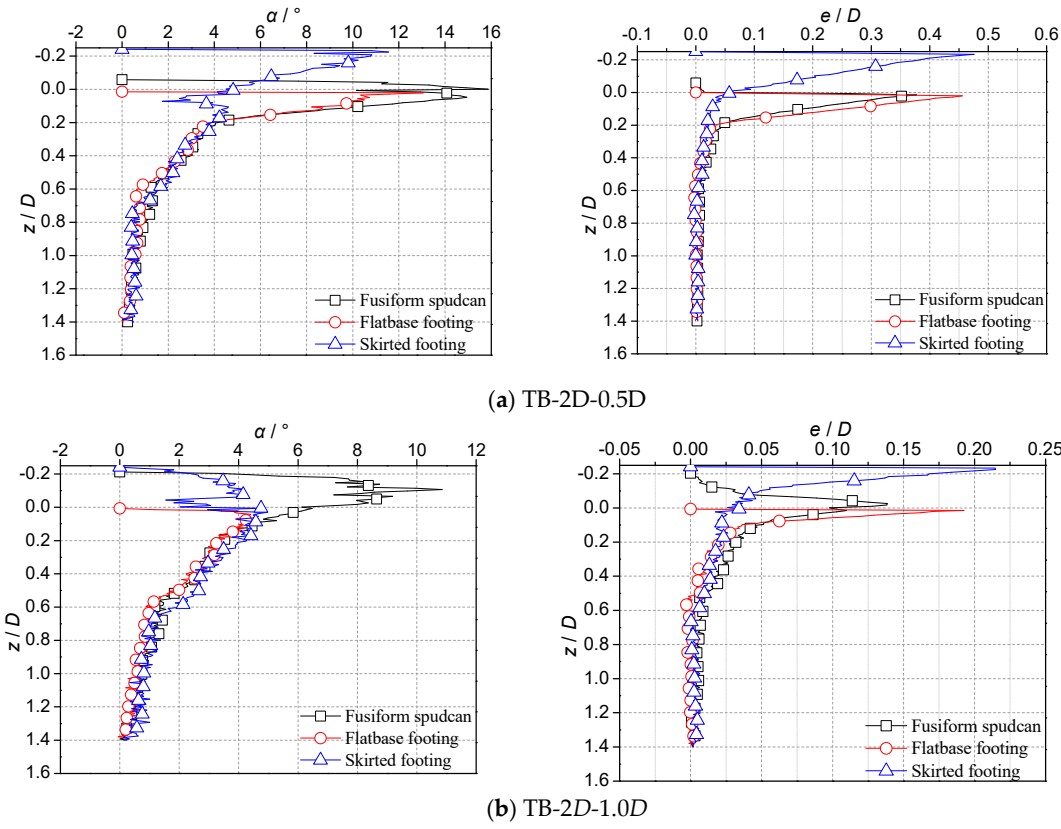

(**a**) TB-2D-0.5D

(**b**) TB-2*D*-1.0*D*

**Figure 21.** Variations in the (**a**) load inclination and (**b**) load eccentricity during the reinstallation process of the β = 0.5*D*, 1.0*D* cases.

## 4. Conclusions

This paper carried out large deformation finite element analyses to investigate the effect of an existing footprint on the stability of jack-ups' reinstallation. The following conclusions can be drawn according to the present numerical analyses:

The friction condition of the soil–footing interface has a significant effect on $H$ profile but much less effect on $M$ profile. The deeper is the footprint, the more effect it has on both $H$ and $M$ profiles.

The eccentricity ratio is a key factor to evaluate $H_{max}$ and $M_{1\text{-}1max}$. For flat base footings and skirted footings, both $H_{max}$ and $M_{1\text{-}1max}$ are significant when $\beta/D = 0.25$ to 1.25. The value of $M_{1\text{-}1max}$ reduces to zero when $\beta/D \geq 1.5$, while $H_{max}$ still remains at a significant value. For fusiform spudcan footings, both $H_{max}$ and $M_{1\text{-}1max}$ are significant when $\beta/D = 0.25$ to 0.5.

The geometry shape of the footing also has a certain effect on the $V$, $H$, and $M$ profiles. The flat base footing gives the lowest $H_{max}$ but the largest $M_{1\text{-}1max}$, and the performances of the fusiform spudcan footing and the skirted footing are similar. From the view of the resultant forces, both $\alpha$ and $e/D$ of the skirted footing are only large before the base level (with the maximum cross-section area) fully touches the soil, which shows a certain potential in resisting the damage during reinstallation near existing footprints by comparing with commonly used fusiform spudcan footings.

The bending moment on the leg–hull connection ($M_{hull}$) at different working leg lengths ($L_{w\text{-leg}}$) is discussed. When $L_{w\text{-leg}}$ is less than ~30 m, the total moment is within a negative range (clockwise). With further increasing of $L_{w\text{-leg}}$, the absolute value of the clockwise $M_{hull}$ would be larger than the anticlockwise $M_{hull}$ at $L_{w\text{-leg}} = 0$.

In this study, the artificial footprints were adopted to simplify the problem neglecting the disturbance of the soil during initial spudcan penetration. In the further study, the soil profiles, soil properties, geometry of footprints and spudcans, leg details, use of spigots (or not) etc. should be noted as a factor to consider in site-specific analyses.

**Author Contributions:** Data curation, X.W.; formal analysis, H.Z.; software, J.L.; supervision, L.Y.

**Funding:** This study was supported by the Chinese National Natural Science Foundation (51890915, 51639002, 51539008 and 51679038).

**Conflicts of Interest:** The authors declare no conflict of interest.

## Appendix A

**Table A1.** The list of peak load.

| Footprint | $\beta/D$ | $H_{max}$/MN | | | $M_{max}$/MN·m | | |
|---|---|---|---|---|---|---|---|
| | | Fusiform Spudcan Footing | Flat Base Footing | Skirted Footing | Fusiform Spudcan Footing | Flat base Footing | Skirted Footing |
| TA | 0.25 | 0.55349 | 0.30883 | 0.42189 | 7.44231 | 9.80965 | 6.09376 |
| | 0.5 | 0.5524 | 0.39296 | 0.6407 | 6.72754 | 10.3384 | 6.84643 |
| | 0.75 | 0.45052 | 0.44328 | 0.68789 | 5.20043 | 9.48458 | 6.57694 |
| | 1.0 | 0.4187 | 0.41402 | 0.6905 | 3.06038 | 11.6203 | 6.06864 |
| | 1.25 | 0.37315 | 0.34509 | 0.62492 | 1.73015 | 8.2809 | 2.8528 |
| | 1.5 | 0.29904 | 0.26207 | 0.51236 | 0.56979 | 1.28662 | 2.54351 |
| TB | 0.25 | 1.0387 | 0.63409 | 0.77274 | 8.59083 | 12.0215 | 6.65431 |
| | 0.5 | 0.99436 | 0.74464 | 1.14443 | 7.42953 | 11.4894 | 7.08853 |
| | 0.75 | 0.8814 | 0.77806 | 1.29267 | 6.00388 | 10.2634 | 7.13538 |
| | 1.0 | 0.80606 | 0.73853 | 1.2015 | 3.99285 | 10.007 | 6.30153 |
| | 1.25 | 0.69096 | 0.64062 | 1.10797 | 1.96356 | 9.37414 | 3.72762 |
| | 1.5 | 0.5602 | 0.56196 | 0.9585 | 0.5096 | 1.77092 | 1.25643 |
| TC | 0.25 | 1.701 | 1.12667 | 1.52282 | 8.93624 | 14.4618 | 7.94377 |
| | 0.5 | 1.78348 | 1.47111 | 2.36442 | 6.0261 | 11.5915 | 7.78829 |
| | 0.75 | 1.57796 | 1.42199 | 2.33573 | 4.96665 | 9.31258 | 6.36396 |
| | 1.0 | 1.50321 | 1.3904 | 2.203 | 4.20818 | 10.5380 | 5.74945 |
| | 1.25 | 1.36363 | 1.25953 | 2.11594 | 1.90344 | 9.36561 | 4.17154 |
| | 1.5 | 1.18654 | 1.09092 | 1.89335 | 1.13969 | 2.79521 | 1.85664 |

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
