# Peer review of "Finite Element Analysis and Parametric Study of Spudcan Footing Geometries Penetrating Clay Near Existing Footprints"

_jmse, doi:10.3390/jmse7060175_

Round 1

Reviewer 1 Report

This paper is generally well-written and illustrated and its main emphasis is on setting out the parametric calculations performed and outlining the results obtained. However, recently, many papers have been published with exactly the same issue (e.g. spudcan penetration nearby footprint) and methodology (e.g. CEL). Seriously, a large portion of the current study is overlapped with those papers. They were not listed in the reference either. 

1.      What is the new contribution of this study? The effect of spudcan geometry on the spudcan-footprint interaction and corresponding soil failure mechanisms have been discussed by many technical papers (see for example Hartono et al., 2014; Jun et al., 2018a and 2018b; 2019).

2.      In the whole paper the author conducted CEL analysis for parametric studies. However, a proper calibration on spudcan-footprint interaction is necessary from either field data or centrifuge tests to prove the reliability of the numerical method.

3.      In the current studies a typical soft soil was used while an artificial footprint profile was cut without properly simulating the soil shear strength variations. This could be another defect of the paper.

4.      No consideration is given of the effects of prior spudcan installation on the soil properties, which can be considerable, especially with sensitive or brittle soils.

5.      The above limitations should be noted and either justified or noted as a factor to consider in site-specific analyses that can take account of the soil profiles, soil properties, geometry of craters and spudcans, leg details, use of spigots (or not) etc.

Author Response

Response to Reviewer 1 Comments

Thank the reviewer very much for the comments. 

Point 1: What is the new contribution of this study? The effect of spudcan geometry on the spudcan-footprint interaction and corresponding soil failure mechanisms have been discussed by many technical papers (see for example Hartono et al., 2014; Jun et al., 2018a and 2018b; 2019).

Response 1: The authors thank the reviewer for the recently published papers. After reading those papers in detail, the authors think the resent paper may still has some contributions, though we focused on similar topics and used similar methods. (Hartono et al., 2014) used experimental method and numerical analysis (simulated with ABAQUS/CEL) respectively to investigate the efficacy of reaming technique in mitigating the footprint hazards. They founded that reaming can be a viable option to mitigate spudcan-footprint interaction. (Jun et al., 2018a; Jun et al., 2018b; Jun et al., 2019; Jun et al., 2018c) conducted some numerical and experimental investigations to explore the efficacy of the novel spudcan in mitigating the footprint hazards. The research object of above studies is fusiform spudcan footing. In this study, the reinstallation of the flat base footing, fusimform spudcan footing and skirted footing near existing footprints are investigated. The effects of footing shape, footprintss geometry, reinstallation eccentricity (0.25D-2.0D) and the roughness between footing and soil on the V H and M profiles are discussed. The bending moments on the leg-hull connection section of different leg length at certain offset distances are also presented. Besides, the discussions on the structure-clay interface friction in ABAQUS/CEL may also be of some values.

Point 2: In the whole paper the author conducted CEL analysis for parametric studies. However, a proper calibration on spudcan-footprint interaction is necessary from either field data or centrifuge tests to prove the reliability of the numerical method.

Response 2: The authors have carried out a further insight to compare the numerical results to the centrifuge test results from (Kong, 2011), and have presented the comparison in the revised paper. It can be seen that the horizontal force profile of Kongs lies between the smooth and rough cases of this study (Figure 1 (a))because the friction characteristic of centrifuge test on the interface of aluminium footing and soil is between rough and smooth. It can be seen that both the numerical results of this study and the centrifuge test results from (Kong, 2011) have very similar H and M profile trends.

(a) Normalized horizontal force

(b) Normalized bending moment

Figure 1 Comparison numerical results (flat base footing) with centrifuge test results after Kong(2011)

Point 3: In the current studies a typical soft soil was used while an artificial footprint profile was cut without properly simulating the soil shear strength variations. This could be another defect of the paper.

Point 4: No consideration is given of the effects of prior spudcan installation on the soil properties, which can be considerable, especially with sensitive or brittle soils.

Response to 3 and 4: The authors agree that the major limitation of this study is neglecting the disturbance of the soil during initial spudcan penetration. To the author’s knowledge, Leung and Gan et al. at the National University of Singapore have studied the spudcan-footprint interaction considering the ‘real’ initial penetration (Gan et al., 2012; Leung et al., 2007; TI, 2009). Their research showed that the soil is obviously disturbed during the initial penetration and will recover with time. To simplify the problem, many of the following researches assumed an artificial footprint, such as (Grammatikopoulou et al., 2007; Jardine et al., 2002; Jun et al., 2018a; Jun et al., 2018b; Jun et al., 2019; Jun et al., 2018c; Kong, 2011; Mao et al., 2015; Zhang, 2015). The assumption of an artificial reverse cone footprint may respond to a fully recovered ‘real’ footprint after a long period from the initial penetration. Therefore, the three idealized footprints following (Kong, 2011) are adopted in this study to simplify the numerical model. The limitations of the study have been added to the CONCLUSIONS in the revised paper.

Point 5: The above limitations should be noted and either justified or noted as a factor to consider in site-specific analyses that can take account of the soil profiles, soil properties, geometry of craters and spudcans, leg details, use of spigots (or not) etc.

Response 5: As presented in the response to 3 and 4, the limitations of the study have been added to the CONCLUSIONS in the revised paper.

Reference

Gan, C.T., Leung, C.F., Cassidy, M.J., Gaudin, C., Chow, Y.K., 2012. Effect of time on spudcan-footprint interaction in clay. Géotechnique 62 (5), 401-413.

Grammatikopoulou, A., Jardine, R., Kovacevic, N., Potts, D., Hoyle, M., Hampson, K., 2007. Potential Solutions To The Problem Of The Eccentric Installation Of Jack-Up Structures Into Old Footprint Craters.

Hartono, H., Tho, K., Leung, C., Chow, Y., 2014. Centrifuge and Numerical Modelling of Reaming as a Mitigation Measure for Spudcan-Footprint Interaction, Offshore Technology Conference-Asia. Offshore Technology Conference.

Jardine, R., Kovacevic, N., Hoyle, M., Sidhu, H.K., Letty, A., 2002. Assessing The Effects On Jack Up Structures Of Eccentric Installation Over Infilled Craters.

Jun, M., Kim, Y., Hossain, M., Cassidy, M., Hu, Y., Park, S., 2018a. Optimising spudcan shape for mitigating horizontal and moment loads induced on a spudcan penetrating near a conical footprint. Applied Ocean Research 79, 62-73.

Jun, M., Kim, Y., Hossain, M., Cassidy, M., Hu, Y., Park, S., 2018b. Physical modelling of reinstallation of a novel spudcan nearby existing footprint, Physical Modelling in Geotechnics, Volume 1. CRC Press, pp. 615-621.

Jun, M., Kim, Y., Hossain, M., Cassidy, M., Hu, Y., Park, S., 2019. Global jack-up rig behaviour next to a footprint. Marine Structures 64, 421-441.

Jun, M., Kim, Y., Hossain, M., Cassidy, M., Hu, Y., Sim, J., 2018c. Numerical investigation of novel spudcan shapes for easing spudcan-footprint interactions. Journal of Geotechnical and Geoenvironmental Engineering 144 (9), 04018055.

Kong, V.W., 2011. Jack-up reinstallation near existing footprints. University of Western Australia.

Leung, C., Gan, C., Chow, Y., 2007. Shear strength changes within jack-up spudcan footprint, The Seventeenth International Offshore and Polar Engineering Conference. International Society of Offshore and Polar Engineers.

Mao, D., Zhang, M., Zhang, L., Duan, M., Song, L., 2015. Sliding risk of jack-up platform re-installation close to existing footprint and its countermeasure. Petroleum Exploration and Development 42 (2), 259-264.

TI, G.C., 2009. Centrifuge model study on spudcan-footprint interaction.

Zhang, W., 2015. 3D Large Deformation Finite Element Analyses of Jack-up Reinstallations Near Idealised

Footprints.

Reviewer 2 Report

Dear authors,

In general the manuscript is well-written.

Some recommendations/suggestions:

The abstract is now more or less a summary of what has been done. However, in the opinion of the reviewer also a 'summary' of the conclusions should be incorporated in the abstract.

It is suggested to change the title into: "Finite Element Analysis and Parametric (Sensitivity) Study of Spudcan Footing Geometries Penetrating Clay Near Existing Footprints.  

In paragraph 2.2 it is mentioned that a cuboid solid domain description is adopted rather than a cylindrical one, since it is easier to mesh and more efficient. Question: are the calculation results the same for both types of domain description?

The manuscript does not include a comparison to experimental data. In the opinion of the reviewer, it means that focus should not be so much on the obtained calculation values itself, but on the comparison of the results for the different footing geometries. The authors did this very well.

Some minor comments:

in the abstract and first part of the manuscript, the terms 'vertical resistance' and 'horizontal resistance' are mentioned together with 'bending moments'. It is suggested to use 'vertical force' and ' horizontal force' together with 'bending moments'.

In figure 4 and the text below the symbol H is introduced, referring to height. However, the horizontal force uses the same symbol. This is confusing.

At row 116 and 117 specific element types are mentioned. It might be an idea to add information about element properties for the used software package (mention that as well).

It is suggested to include 'footing' for all geometries: flat base footing, fusiform spudcan footing and skirted footing.

Concerning the writing, it is suggested to look one more time very carefull. Then the authors can solve minor issues like: changes the seabed (row 34); an inclined seabed surface and a varying soil strength profile (row 34); at a rate (row 40); V, H and M (row 81).

Some figure references are missing. For example at row 242.

At row 90 and row 126 a reference is missing.

Author Response

Response to Reviewer 2 Comments

Point 1:The abstract is now more or less a summary of what has been done. However, in the opinion of the reviewer also a 'summary' of the conclusions should be incorporated in the abstract.

Thank the reviewer very much for the comments.

Response 1: The modified abstract:

Most existing researches on the stability of spudcans during reinstallation nearing footprints are from centrifuge tests and theoretical analyses. In this study, the reinstallation of the flat base footing, fusimform spudcan footing and skirted footing near existing footprints are simulated using Coupled Eulerian - Lagrangian (CEL) method. The effects of footprints geometry, reinstallation eccentricity (0.25D-2.0D) and the roughness between spudcan and soil on the profiles of the vertical force, horizontal force and bending moment are discussed. The results show that the friction condition of the soil-footing interface has great effect on H profile but much less effect on M profile. The eccentricity ratio is a key factor to evaluate the H and M. The results show that the geometry shape of the footing also has certain effects on the V H and M profiles. The flat base footing gives the lowest peak value in H but largest in M, and the performances of the fusiform spudcan footing and the skirted footing are similar. From the view of the resultant forces, the skirted footing shows a certain potential in resisting the damage during reinstallation near existing footprints by comparing with commonly used fusiform spudcan footings. The bending moments on the leg-hull connection section of different leg length at certain offset distances are discussed.

Point 2: It is suggested to change the title into: "Finite Element Analysis and Parametric (Sensitivity) Study of Spudcan Footing Geometries Penetrating Clay Near Existing Footprints.

Response 2: The title of this paper has been changed into:Finite Element Analysis and Parametric Study of Spudcan Footing Geometries Penetrating Clay near Existing Footprints.

Point 3: In paragraph 2.2 it is mentioned that a cuboid solid domain description is adopted rather than a cylindrical one, since it is easier to mesh and more efficient. Question: are the calculation results the same for both types of domain description?

Response 3: Yes, they are the same. The authors have carried out the sensitivity study on the mesh. As shown in Figure 1, a fusiform spudcan footing is installed in cylindrical soil field model and cuboid soil field model respectively with v = 0.5 m/s. All model parameters remain the same except the soil model shape. It can been seen from Table 1 that the calculation results are extremely close, but the time cost of cuboid soil model case is much less. It may be because that the elements under the footing in the cuboid model have better aspect ratio.

(a) Cylindrical soil model

(b) Cuboid soil model

Figure 1 Two types of soil model

Table 1 Comparison between cylindrical and cuboid soil model

Soil Model

Su (kPa)

Model parameters

Nc

Error

Time cost

Cylinder

10

Penetration velocity v = 0.5 m/s

Penetration depth z/D=2.1

Submerged unit weight of soil γ’ = 10

13.50

0.7%

192min

Cuboid

13.40

91min

Cylinder

39

12.99

0.3%

158min

Cuboid

12.95

110min

Point 4:The manuscript does not include a comparison to experimental data. In the opinion of the reviewer, it means that focus should not be so much on the obtained calculation values itself, but on the comparison of the results for the different footing geometries. The authors did this very well.

Response 4: Thank you for your suggestion. The authors have carried out a further insight to compare the numerical results to the centrifuge test results from (Kong, 2011), and have presented the comparison in the revised paper. It can be seen that the horizontal force profile of Kongs lies between the smooth and rough cases of this study (Figure 2 (a))because the friction characteristic of centrifuge test on the interface of aluminium footing and soil is between rough and smooth. It can be seen that both the numerical results of this study and the centrifuge test results from (Kong, 2011) have very similar H and M profile trends.

(a) Normalized horizontal force

(b) Normalized bending moment

Figure 2 Comparison numerical results (flat base footing) with centrifuge test results after Kong(2011)

Point 5: In the abstract and first part of the manuscript, the terms 'vertical resistance' and 'horizontal resistance' are mentioned together with 'bending moments'. It is suggested to use 'vertical force' and ' horizontal force' together with 'bending moments'.

Response 5: Revised throughout the paper.

Point 6: In figure 4 and the text below the symbol H is introduced, referring to height. However, the horizontal force uses the same symbol. This is confusing.

Response 6: The symbol of the height has been changed into:Ha

Figure 4. The dimensions of footings: flat base footing, fusiform spudcan and skirted footing

Line 114-115: The distance from centre of section 1-1 to reference point(RP) for flat base footing is Ha =1.75 m, for fusiform spudcan is Ha =7.55 m, for skirted footing is Ha =5.5 m.

Point 7: At row 116 and 117 specific element types are mentioned. It might be an idea to add information about element properties for the used software package (mention that as well).

Response 3: The commonly used element type in ABAQUS/CEL are taken. The detailed information of the element and software package are stressed in the revised paper as “The soil is modeled by EC3D8R element (three-dimensional, 8-node linear brick, multimaterial, reduced integration with hourglass control) and the footings are modeled by C3D8R (three-dimensional, 8-node linear brick, reduced integration with hourglass control) element in ABAQUS/Explicit. 

Point 7: It is suggested to include 'footing' for all geometries: flat base footing, fusiform spudcan footing and skirted footing.

Response 4: The name of fusiform spudcan has been collectively referred to as fusiform spudcan footing.

Point 8: Concerning the writing, it is suggested to look one more time very carefull. Then the authors can solve minor issues like: changes the seabed (row 34); an inclined seabed surface and a varying soil strength profile (row 34); at a rate (row 40); V, H and M (row 81).

Response 8: Revised as suggested. The writing throughout the paper has been checked again carefully.

Point 9: Some figure references are missing. For example at row 242. At row 90 and row 126 a reference is missing.

Response 9: I have revised these mistakes, thanks again for your suggestion. Please allow me to explain that the missing of figure references might due to the conflicts between different office products because the figure references are well cited in the doc which I uploaded before.  

Round 2

Reviewer 1 Report

Authors addressed all comments/suggestions pointed by the reviewer. Therefore, I recommend the publication of the work.